# Curcumin Alleviates HMGB1-Mediated Inflammation Through the Signaling Pathway of *TLR2*-NF-κB in Bovine Ovarian Granulosa Cells

**DOI:** 10.3390/ijms26189180

**Published:** 2025-09-19

**Authors:** Siqi Liu, Yingying Xie, Lei Wang, Jingyan Zhang, Xiaoliang Chen, Xiaowei Feng, Junyan Wang, Kang Zhang, Jianxi Li

**Affiliations:** 1Technology Innovation Center of Traditional Chinese Veterinary Medicine of Gansu Province, Lanzhou Institute of Husbandry and Pharmaceutical Sciences of CAAS, Lanzhou 730050, China; liusiqi231004@163.com (S.L.); xieying_vet@163.com (Y.X.); wanglei03@caas.cn (L.W.); zwzh1223@126.com (J.Z.); 18148345073@163.com (X.C.); 82101231377@caas.cn (X.F.); wjy620422@163.com (J.W.); 2College of Animal Science and Technology, Gansu Agricultural University, Lanzhou 730070, China; 3College of Animal Science and Veterinary Medicine, Heilongjiang Bayi Agricultural University, Daqing 163000, China

**Keywords:** curcumin, HMGB1, bovine ovarian granulosa cells, *TLR2*, inflammation

## Abstract

Curcumin, the principal bioactive compound derived from turmeric, possesses a wide range of therapeutic properties such as anti-inflammatory, antioxidant, and wound-healing properties. Recent studies suggest that curcumin may alleviate HMGB1-mediated inflammation in ovarian cells. However, its role in modulating dysfunction in HMGB1-driven ovarian granulosa cells (OGCs) remains to be elucidated. In the present study, curcumin suppresses the HMGB1-induced overexpression of toll-like receptor 2 (*TLR2*) and ovulation-related factors such as *EGFR*, *VEGF*, *STAR*, and *TIMP1/2* genes. Additionally, the elevated levels of *TLR2*, *TLR1*, *TLR6*, and phospho-NF-κB p65 proteins were significantly inhibited by curcumin. Further mechanistic analysis reveals that the interaction between HMGB1 and the *TLR1*-*TLR2*/*TLR6* complex, as well as phospho-NF-κB p65, was restrained. This resulted in the suppression of the pro-inflammatory cytokine IL-6 production and the alleviation of the HMGB1-induced inflammation response in OGCs. Collectively, our findings demonstrate that curcumin modulates the upregulation of ovulation-related genes and pro-inflammatory cytokines in OGCs by inhibiting the *TLR2*-NF-κB pathway, providing a mechanistic basis for its potential application as a therapeutic agent against OGC inflammation.

## 1. Introduction

Turmeric (*Curcuma longa* L.), a perennial rhizomatous herb, is globally recognized for its dual roles as a culinary spice and a medicinal agent [1]. In traditional Chinese medicine, turmeric has been utilized to improve human or animal reproductive efficiency [2]. For instance, some Chinese herbal compounds containing turmeric promote ovarian cycle recovery, improve follicle development, and enhance immunological and uterine cleansing capacity in postpartum buffaloes [3,4]. Curcumin, as the earliest discovered and most representative linear diarylheptanoid compound, has been extensively researched for its extensive anti-inflammatory, antioxidant, and anti-tumor properties [5,6], as well as its potential in neuroprotection, antiviral applications, and the treatment of ovarian disorders. Rodent studies have revealed that curcumin promotes folliculogenesis, prevents premature ovarian failure, and enhances fertility by suppressing oxidative stress and inflammation [7,8]. Recent studies have indicated that curcumin protects H9C2 cardiomyocytes in rats against hypoxia/reoxygenation injury, demonstrating a biphasic effect [9]. In an investigation examining the cytotoxicity and alkaline phosphatase activity of curcumin, aloin, and MTA on human dental pulp cells, curcumin at 2.5 μg/mL significantly promoted cell proliferation on day 1 [10]. Furthermore, treatment with curcumin at concentrations of 5, 10, and 20 μM was shown to significantly enhance anti-inflammatory effects in mouse fibroblasts [11]. Based on these findings, concentrations of 0, 1.25, 2.5, 5, 10, 20, and 40 μg/mL were selected to establish an effective and non-cytotoxic range in bovine ovarian granulosa cells (BOGCs). This gradient enables a comprehensive evaluation of the dose-dependent therapeutic potential of curcumin. Mechanistically, curcumin exerts anti-inflammatory, anti-angiogenic, and anti-cancer effects in ovarian tissues, which are primarily attributed to its inhibitory action on the nuclear transcription factor NF-κB [12,13]. For example, ST09, a curcumin derivative, can regulate cell proliferation and migration through the miR-199a-5p/DDR1 axis in ovarian cancer cells, which is associated with the inhibition of the NF-κB signaling pathway [14]. Despite these advances, the molecular interplay between curcumin and key inflammatory mediators in ovarian physiology remains poorly understood, particularly its interaction with the HMGB1-*TLR2* signaling pathway in BOGCs.

High-mobility group box 1 (HMGB1) is a nuclear protein that plays an important regulatory role in DNA replication, transcription, and repair. In pathogenic situations, HMGB1 is actively secreted by immune cells or passively produced during cellular apoptosis, acting as a damage-associated molecular pattern (DAMP) [15,16]. HMGB1 fortifies chromatin architecture and oversees DNA repair processes within the nucleus, while extracellular HMGB1 initiates pro-inflammatory cascades through interaction with receptors such as Toll-like receptor 2 (*TLR2*), TLR4, and the receptor for advanced glycation end products (RAGE) [15,17]. It was shown that HMGB1 is implicated in placental inflammation, endometriosis, and polycystic ovarian syndrome (PCOS) [18,19]. Transcriptomic analyses have revealed the increase of HMGB1 expression in terminally differentiated OGCs from aging ovaries, suggesting its role in ovarian aging and dysfunction [20]. Our previous studies have demonstrated that HMGB1 is involved in ovarian innate immunity and follicle maturation by modulating the expression of critical genes such as *EGFR*, *VEGF*, *STAR*, and *TIMP1/2* via the *TLR2*-NF-κB pathway in BOGCs [21,22].

Within the ovarian follicle, a delicate balance between pro-inflammatory and anti-inflammatory factors is crucial for normal follicular growth and ovulation [23]. Disruption of this balance, which is mediated by excessive HMGB1 release, can lead to impaired follicular development and anovulation [24]. HMGB1 exerts pro-inflammatory effects by amplifying the *TLR2*-mediated signaling pathway, which disrupts follicular homeostasis and promotes pathogenic processes [17]. The multi-target anti-inflammatory properties of curcumin have demonstrated potential in attenuating HMGB1-driven inflammation in various disease models [25]. However, the specific mechanisms by which curcumin may affect BOGCs by interfering with the HMGB1-*TLR2*-NF-κB signaling pathway remain to be elucidated.

The present study aims to investigate the molecular mechanisms underlying curcumin-mediated alleviation of HMGB1-induced inflammation and regulation of ovulation-related genes in BOGCs. The results demonstrated that curcumin shows activity to regulate the upregulation of ovulation-related genes and pro-inflammatory cytokines in OGCs, which is associated with the inhibition of the *TLR2*-NF-κB pathway. The findings provide fresh perspectives on the therapeutic potential of curcumin for ovarian inflammation.

## 2. Results

### 2.1. Biphasic Modulation of BOGCs Viability by Curcumin

Recent evidence indicates that the pro- and anti-cell viability effects of curcumin are controversial [9,26,27]. To determine the therapeutic window of curcumin in BOGCs, the dose-dependent effects on cellular viability were primarily evaluated using the MTT assay. The result indicated that 5 μg/mL curcumin significantly increased granular cell viability, whereas concentrations ≥ 20 μg/mL curcumin decreased granular cell viability (Figure 1). The cell viability was not significantly altered by the concentration at 1.25, 2.5, 5, and 10 μg/mL (Figure 1).

### 2.2. Curcumin Demonstrates Dose-Dependent Regulation of Apoptosis

Fluorescence microscopy coupled with Annexin V/PI dual staining revealed a dichotomous effect of curcumin survival on BOGCs. Low-dose curcumin (1.25–5 μg/mL) reduced apoptosis, while high-dose curcumin (20–40 μg/mL) exacerbated cell death (Figure 2A–C). Quantitative analysis confirmed 5 μg/mL curcumin suppressed early apoptosis and late apoptosis/necrosis compared to untreated controls (0 μg/mL) (Figure 2B,C).

### 2.3. Curcumin Reverses HMGB1-Driven TLR2-Associated Ovulatory Gene Upregulation

HMGB1 challenge (5 μg/mL) significantly upregulated *TLR2* and *TLR2*-associated ovulatory genes, such as *EGFR*, *VEGF*, *STAR*, *TIMP1*, and *TIMP2* (Figure 3A–F). For *TLR2*, *EGFR,* and *VEGF,* curcumin attenuated these effects in a dose-dependent manner, whereas the optimal inhibitory concentrations of curcumin were 2.5 μg/mL, 1.25 μg/mL, and 5 μg/mL for *STAR*, *TIMP1,* and *TIMP2*, respectively (Figure 3A–D). Curcumin exhibited markedly inhibitory effects compared to siRNA-*TLR2* in suppressing *STAR*, *VEGF*, and *TIMP1* (Figure 3C–E). In contrast, *TIMP2* remained unresponsive to both interventions (Figure 3F).

### 2.4. Curcumin Inhibits the HMGB1-TLR2/NF-κB Signaling Axis

Western blot analysis demonstrated curcumin reduced the levels of *TLR1*, *TLR2*, *TLR6*, and phospho-NF-κB p65 proteins in a dose-dependent manner (Figure 4A). At the concentration of 5 μg/mL, curcumin suppressed *TLR2* expression and NF-κB phosphorylation to an extent comparable to that of siRNA-*TLR2* (Figure 4B–E). These results showed that curcumin inhibits HMGB1-induced *TLR1*-*TLR2*/6 heterodimers and phospho-NF-κB p65.

### 2.5. Curcumin Inhibits HMGB1-Induced NF-κB Nuclear Translocation

Essential mechanisms that mediate inflammatory responses encompass NF-κB signaling and nuclear translocation [28]. To determine the impact of curcumin on the nuclear translocation of the NF-κB p65 subunit in granulosa cells treated by HMGB1, an indirect immunofluorescence assay was used to detect the localization of the NF-κB p65 subunit. The results indicated that both curcumin and siRNA-*TLR2* attenuated HMGB1-induced nuclear translocation of NF-κB p65 (Figure 5A). Immunofluorescence quantification revealed that HMGB1-induced nuclear p65 accumulation was significantly decreased when the dose of curcumin was 5 μg/mL (Figure 5D). *TLR2* is the primary mediator of HMGB1 signaling, as siRNA-*TLR2* recapitulated this effect (Figure 5B–D).

### 2.6. Curcumin Attenuates the Binding of HMGB1 to TLR1, TLR2, TLR6 and Phospho-NF-κB p65

Co-IP assays demonstrated that curcumin directly disrupts the interaction between HMGB1 and its receptors (Figure 6A,C,E,G). Curcumin at 5 μg/mL significantly reduced HMGB1 binding to *TLR1*, *TLR2*, *TLR6*, and phospho-NF-κB p65 (Figure 6B,D,F,H). Notably, the structural interference of curcumin at 5 μg/mL on *TLR6* exceeded the inhibitory effects of siRNA-*TLR2* (Figure 6F).

### 2.7. Curcumin Reduces HMGB1-Induced Increase of the Inflammatory Factor IL-6

Interleukin 6 (IL-6) and Tumor Necrosis Factor α (TNF-α) are significant inflammatory cytokines in cells [29]. ELISA analysis showed that the ability of 5 μg/mL curcumin to reduce HMGB1-induced IL-6 secretion is more than that of siRNA-*TLR2* (Figure 7A). However, TNF-α levels are not changed (Figure 7B), indicating the specificity of the *TLR2*/NF-κB-IL-6 axis.

## 3. Discussion

Curcumin, a natural polyphenolic molecule serving as both a food additive and pharmacological agent, has demonstrated remarkable anti-inflammatory properties through multi-target modulation [30]. Although its therapeutic potential in metabolic and tumor diseases has been well documented [31,32,33], its role in ovarian pathophysiology processes remains fully explored. Ovarian inflammation is a significant contributor to reproductive failure in cows, leading to substantial reductions in both fertility and overall reproductive performance [34]. Currently, the anti-inflammatory and antioxidant activities of curcumin have been extensively investigated [35]. Studies have indicated that curcumin amplifies the anti-inflammatory properties of Tehranolide via modulating the STAT3/NF-κB signaling cascade in ovarian cancer cell lines [12]. Our present study indicates that curcumin can improve the abnormal changes of inflammatory factors and ovulation-related genes by acting as a modulator of the HMGB1-*TLR2*-NF-κB axis in BOGCs, thereby suggesting its potential involvement in fertility control pathways; however, functional validation in a reproductive model is necessary to substantiate this claim.

Physiological ovulation involves a strictly regulated inflammatory cascade mediated by prostaglandins and those cytokines related to reproduction [36,37]. As observed in polycystic ovary syndrome, the follicular homeostasis was disrupted by pathological overactivation of this process through the accumulation of excessive DAMPs [38]. Among these DAMPs, HMGB1 has emerged as a key inflammatory mediator across various tissues. For example, it promotes tumorigenesis in colitis-associated colorectal cancer via the ERK1/2 pathway [39], and in collagen-induced arthritis, inhibition of the HMGB1/TLR4/STAT3 axis in M1 macrophages alleviates disease severity [40]. In the context of the ovary, HMGB1 has been linked to polycystic ovary syndrome, chronic inflammation, and miscarriage [19]. In our study, we observed that HMGB1 triggers *TLR2*-dependent NF-κB activation, upregulates IL-6, and disrupts the expression of ovulation-related genes (*EGFR*, *VEGF*, *STAR*, and *TIMP1/2*), positioning it as a central disruptor of BOGC function and a contributor to ovarian inflammatory microenvironments. This discovery establishes HMGB1/*TLR2* signaling as a gatekeeper between physiological ovulation and inflammatory ovarian pathology. Moreover, the abnormal increase of *EGFR*, *VEGF*, *STAR*, and *TIMP1/2* expression may indirectly lead to ovarian microenvironment disorder and ovarian inflammation. Our data further illustrate that HMGB1/*TLR2* signaling may play a crucial role in the transition from healthy ovulation to ovarian inflammatory disease. Notably, curcumin exhibited a greater inhibitory effect on the expression of *STAR*, *VEGF*, and *TIMP1* than did siRNA-*TLR2*, suggesting the existence of additional *TLR2*-independent mechanisms (Figure 3C–E).

As the principal bioactive constituent of turmeric, curcumin is the core substance through which turmeric exerts its anti-inflammatory effects, and there is 2–5% content in the turmeric rhizome [41]. Owing to its potent anti-inflammatory properties, a novel nanobiological preparation derived from turmeric-derived nanovesicles has been developed for the targeted treatment of ulcerative colitis, with the aim of restoring the compromised intestinal barrier [42]. Numerous studies have demonstrated that curcumin reduces the expression of inflammatory factors, including IL-1β, IL-6, IL-8, and TNF-α, by regulating NF-κB signaling [43,44]. In the current study, curcumin exhibits a biphasic regulatory activity on ovarian granulosa cells, which is characterized by low-dose promotion and high-dose inhibition of cellular activity. At optimal concentrations (≤5 μg/mL), curcumin suppressed *TLR2* expression at transcriptional and translational levels, thereby effectively blocking HMGB1-induced NF-κB activation and subsequently preventing overproduction of IL-6. Crucially, curcumin restores HMGB1-driven dysregulation of ovulatory-related genes (*TIMP1/2*, *STAR*), suggesting its pleiotropic mechanisms extend anti-inflammation actions—potentially involving epigenetic modulation or post-translational modifications of key ovulatory mediators. This multi-target pharmacological profile differentiates curcumin from conventional TLR antagonists, which typically exert inhibitory effects on single signaling pathways [45].

Interestingly, our data suggest a potential direct interaction between HMGB1 and curcumin, which was proved by the reduction of HMGB1-*TLR2* complex formation following curcumin treatment. This finding is consistent with emerging evidence demonstrating curcumin’s structural ability to neutralize DAMPs [46], proposing a dual mechanism involving *TLR2* expression downregulation and HMGB1 ligand inactivation. This dual inhibitory action may synergistically protect BOGCs from inflammatory overload, potentially offering superior efficacy compared to single-target therapeutic strategies.

However, curcumin’s narrow therapeutic window in BOGCs, as indicated by cytotoxicity at concentrations exceeding 5 μg/mL, highlights significant formulation challenges. Previous studies have found that curcumin has very poor water solubility at physiological pH [47]. Additionally, it undergoes extensive phase II metabolism while passing through the intestinal brush border membrane, primarily catalyzed by the UGT1A8 and UGT1A10 enzymes in the intestine, resulting in glucuronidation that produces curcumin glucuronide [48,49,50]. This indicates that the majority of curcumin is digested before entering the bloodstream, resulting in small amounts of the original compound reaching the circulation. Moreover, reductase modifies the olefin bond of curcumin, transforming it into tetrahydrocurcumin (THC) and diminishing its biological activity [50]. Curcumin has a brief half-life in the body due to its quick metabolism. The half-life in rat tissues varies from 12.6 to 48.8 min, whereas the average residence duration spans from 20.4 to 75.7 min [51]. Although its low bioavailability has historically limited clinical translation [44,52], recent advances in nanoparticle-based encapsulation technologies offer promising strategies to enhance ovarian targeting while reducing systemic toxicity [53].

Nanostructured lipid carriers have a high encapsulation efficiency and drug loading capacity, making them ideal for the prolonged release of curcumin [44,54]. However, the clinical or veterinary application of the curcumin nano-delivery system still faces the realistic challenges of a complex process, physical and chemical stability to be optimized, and high industrial production cost.

## 4. Materials and Methods

### 4.1. Primary Reagents and Consumables

MTT Solution (M1025) was acquired from Solarbio, Beijing, China. Curcumin (C1386) and DMSO (D2653) were acquired from Sigma, St. Louis, MO, USA. *TLR2* (bs-1019R), NF-κB p65 (bs-0465R), and phospho-NF-κB p65 (bs-0982R) antibodies were acquired from Bioss, Beijing, China. *TLR1* (orb48968) and *TLR6* (orb357127) antibodies were acquired from Biorbyt, Cambridge, UK. β-actin monoclonal antibody (66009-1-Ig), HMGB1 monoclonal antibody (66525-1-Ig) and CoraLite488-conjugated Goat Anti-Rabbit IgG (SA00013-2) were acquired from Proteintech, Rosemont, IL, USA. FITC-Annexin V and PI Apoptosis Kit (C1062M), PVDF membrane (FFP39), BeyoGel™ Plus Precast PAGE Gel (P0456M), SDS-PAGE protein loading buffer (P0015), BCA Protein Assay Kit (P0012), SDS-PAGE Electrophoresis Buffer (Tris-Gly, Powder, P0014B), Western Transfer Buffer (P0021B) and Blocking Buffer (P0023B) were acquired from Beyotime, Shanghai, China. The high-purity RNA rapid extraction kit (RP1202) was acquired from Bioteke, Beijing, China. The reverse transcription kit (RR037A) and TB Green Premix Ex Taq II (RR820A) were acquired from Takara, Kusatsu, Shiga, Japan. M-PER™ Mammalian Protein Extraction Reagent (78501), Pierce™ Classic IP kit (26146), and Restore™ PLUS Western Blot Stripping Buffer (46430) were acquired from Thermo Scientific, Waltham, MA, USA. WesternBright Sirius HRP substrate (K12043-D20) was acquired from Advansta, San Jose, CA, USA. HMGB1 (ab82100), TNF-α (ab193683), and the IL-6 ELISA kit (ab205080) were acquired from Abcam, Cambridge, UK. *TLR2*-siRNA was acquired from HANBIO, Shanghai, China. Cell strainers (352340, 352350) were acquired from Falcon, Miami, FL, USA. Cell culture bottles (430639, 431080), cell culture plates (3516, 3799, 3526), and centrifuge tubes (430790, 430828) were acquired from Corning, Corning, NY, USA. RNase-free & DNase centrifuge tubes (MCT-150-C, PCR-02-C) and 8-strip tubes (PCR-0208-C) were acquired from Axygen, Union City, CA, USA.

### 4.2. Instruments and Equipment

Real-time fluorescence quantitative PCR instrument (Thermo Fisher Scientific, ABI Q5, Waltham, MA, USA); High speed refrigerated centrifuge (Eppendorf, ThermoMixer C, Hamburg, Germany); SDS-PAGE electrophoresis apparatus (Bio-Rad, PowerPac, Hercules, CA, USA); Cell incubator (Heal Force, HF90, Shanghai, China); GeneGnome XRQ Chemiluminescence lmaging System (Gene Company Limited, GGNOME-XRQ-NP, Hong Kong, China); Intelligent microplate reader (Bio Tek, SYNERGYILX, Santa Clara, CA, USA); Micro volume spectrotometer (Life Real, NanoReady, Hangzhou, China); Inverted fluorescence microscope (Olympus, IX71, Tokyo, Japan); General optical microscope (Olympus, BX43); Medical cryostorage refrigerator (Haier, DW-86L490J, Qingdao, China); Laser confocal scanning microscope (Carl Zeiss AG, ZISS LSM-800, Oberkochen, Germany).

### 4.3. Isolation and Culture of BOGCs

Two hundred pairs of ovaries from 100 healthy Holstein dairy cows were procured from Ningxia Luoheqiao Meat Food Co., Ltd., located in Wuzhong City, Ningxia Hui Autonomous Region, China. The ovaries were immersed in normal saline (containing 2% penicillin and streptomycin) at 37 °C and transported to the laboratory. Surplus blood arteries and connective tissue were removed using surgical scissors. The ovaries were washed twice with 75% alcohol and three times with PBS (containing 2% penicillin and streptomycin). The BOGCs were then collected using a 5 mL sterile syringe and placed into a 15 mL centrifuge tube. The cells were rinsed three times with PBS (containing 2% penicillin and streptomycin). Following centrifugation, the isolated BOGCs were inoculated into a T75 culture flask.

### 4.4. MTT Assay

BOGCs were inoculated into 96-well plates, and curcumin concentrations of 0 μg/mL, 0.625 μg/mL, 1.25 μg/mL, 2.5 μg/mL, 5 μg/mL, 10 μg/mL, 20 μg/mL, and 40 μg/mL were then added, followed by a 24 h incubation period. Subsequently, 10 μL of 5 mg/mL MTT solution was introduced into each well, and the plates were incubated at 37 °C in a 5% CO_2_ incubator for 4 h, shielded from light. Following the removal of the supernatant, 150 μL of DMSO was introduced to each well and incubated at 37 °C for 10 min. The optical density was assessed at 490 nm.

### 4.5. Apoptosis Detection

BOGCs were infected on 24-well plates and subjected to curcumin treatment for 24 h. The cell culture medium was discarded, and the cells were rinsed with PBS. Subsequently, 195 μL of FITC-Annexin V binding solution and 5 μL of Annexin V-FITC were included and gently mixed. Subsequently, 10 μL of propidium iodide (PI) staining solution was added and mixed gently. The plates were incubated at room temperature for 10–20 min in the dark and then placed in an ice bath. FITC-Annexin V fluorescence (green) and PI fluorescence (red) were observed under a fluorescence microscope.

### 4.6. Cell Treatment

BOGCs were infected in 6-well plates, and HMGB1 and curcumin were diluted in DMEM/F12 (1:1) medium containing 10% FBS. siRNA-*TLR2* transfection was performed when BOGCs reached 70% confluency. After transfection for 24 h, HMGB1 and curcumin were added. The samples were collected after continuous culture for 24 h. The treatment groups were as follows: 0 μg/mL HMGB1, 5 μg/mL HMGB1, 5 μg/mL HMGB1 + 1.25 μg/mL curcumin, 5 μg/mL HMGB1 + 2.5 μg/mL curcumin, 5 μg/mL HMGB1 + 5 μg/mL curcumin, 5 μg/mL HMGB1 + siRNA-*TLR2*, and siRNA-*TLR2* alone.

### 4.7. Total RNA Extraction and cDNA Synthesis from BOGCs

The growth medium was removed, and 1 mL of RL lysate was added to extract total RNA using the manufacturer’s instructions for the High Purity RNA Rapid Extraction Kit. The quality and amount of RNA purity were evaluated using a microvolume spectrophotometer. A reverse transcription kit was used to reverse transcribe 2 μg of total RNA in a 20 μL reaction volume. The resultant cDNA was preserved at −20 °C.

### 4.8. Reverse Transcription-Quantitative PCR (RT-qPCR)

RT-qPCR was conducted using a two-step protocol on the ABI Q5 Real-Time Fluorescence Quantitative PCR apparatus. The reaction conditions included predenaturation at 95 °C for 30 s, followed by 40 cycles of 95 °C for 5 s and 60 °C for 30 s. The reaction system (20 μL) included 12.5 μL of TB Green Premix Ex Taq II, 0.8 μL of Forward Primer (10 μM), 0.8 μL of Reverse Primer (10 μM), 1 μL of cDNA template, and 7.4 μL of RNase-free ddH_2_O. The primers used in the system were designed and synthesized by TaKaRa Bio Biological Company, and their sequences are listed in Table 1. The reaction was carried out in 8-strip tubes, with each sample analyzed in triplicate. The internal reference gene was *ACTB*. The relative expression of target genes was determined using the 2^−△△Ct^ technique.

### 4.9. Western Blot (WB)

BOGCs were rinsed with pre-chilled PBS. Protease and Phosphatase Inhibitor Cocktail was added to M-PER Mammalian Protein Extraction Reagent. Then, 100 μL of the protein extraction reagent was introduced into each well of a 6-well plate to acquire total protein. The protein content was assessed with the BCA Protein Assay Kit, and the samples were equilibrated and denatured with 5X SDS-PAGE protein loading buffer. SDS-PAGE electrophoresis was conducted at 150 V until complete separation of the protein markers was achieved. The proteins were transferred to a PVDF membrane at 360 mA for 60 min. The membrane was then blocked using the Blocking Buffer after the transfer. Primary antibodies targeting *TLR1*, *TLR2*, *TLR6*, phospho-NF-κB p65, NF-κB p65, and ACTB were incubated at 4 °C overnight. The membrane underwent three washes with TBST, then was incubated with the secondary antibody at room temperature for 1 h with moderate agitation (80 rpm). Following three washes with TBST, WesternBright Sirius HRP substrate was used for development, and pictures were captured using chemiluminescence imaging equipment. The gray values of the bands were examined using ImageJ v1.53q software.

### 4.10. Indirect Immunofluorescence (IIF)

The media was removed from the 6-well plates, and the cells were delicately rinsed three times with pre-chilled PBS. Subsequently, 200 μL of tissue fixative was introduced to each well and incubated at room temperature for 25 min with moderate agitation. The cells underwent three washes with PBS, each lasting 5 min. Subsequently, 200 μL of 0.1% Triton-X-100 was administered for permeabilization at ambient temperature for 10 min, followed by three washes with PBS. Subsequently, 200 μL of 10% goat serum was introduced to each well and incubated at room temperature for 50 min with moderate agitation. Subsequent to the removal of the goat serum, the NF-κB p65 antibody was promptly introduced and incubated at 4 °C overnight. The next day, the plates were incubated at ambient temperature for 30 min and washed three times with PBS. CoraLite488-conjugated Goat Anti-Rabbit IgG was administered and incubated at 37 °C for 1 h. Following three washes with PBS, the cells were stained with DAPI for 7 min at room temperature in the dark and then rinsed three times with PBS. The target protein’s expression was examined using an inverted fluorescence microscope, and photos were captured.

### 4.11. Co-Immunoprecipitation (CO-IP)

The cell culture medium was removed, and the cells were rinsed with pre-chilled PBS. Pre-cooled IP lysis/wash buffer (300 μL) was added to each well and incubated on ice for 5 min, with regular mixing to ensure complete cell lysis. The lysate containing cellular debris was transferred to a microcentrifuge tube and subjected to centrifugation at 13,000× *g* for 10 min. Then, 80 μL of control agarose resin slurry was introduced to a centrifuge column and centrifuged at 1000× *g* for 1 min to remove the storage buffer. Next, 100 μL of 0.1 M Na_3_PO_4_, 0.15 M NaCl, pH 7.2 buffer was introduced to the column and centrifuged at 1000× *g* for 1 min, and the flow-through was discarded. Then, 1 mg of cell lysate was added to the column containing the resin and incubated at 4 °C for 1 h with moderate agitation. After centrifugation at 1000× *g* for 1 min, the column was discarded, and the flow-through was retained. Then, 8 μg of HMGB1 antibody was mixed with the pretreated cell lysate in a microcentrifuge tube. The antibody/cell lysate mixture was diluted with IP lysis/wash buffer to a final volume of 300 μL and incubated overnight at 4 °C to form immune complexes. Pierce Protein A/G agarose resin was gently homogenized, and 20 μL of the resin slurry was introduced onto a Pierce centrifugal column. The column was positioned in a microcentrifuge tube and centrifuged at 1000× *g* for 1 min, after which the flow-through was discarded. The antibody/cell lysate samples were added to the column containing protein A/G agarose resin and incubated for 1 h with gentle shaking. After centrifugation at 1000× *g* for 1 min, the flow-through was discarded. The column was placed in a new collection tube, and the resin was washed twice with 200 μL of pre-cooled IP lysis/wash buffer and once with 100 μL of conditioning buffer (1X). Finally, 100 μL of SDS-PAGE protein loading buffer (2X) was included, and the sample was subjected to heating at 100 °C for 8 min. The sample was cooled to room temperature before SDS-PAGE electrophoresis.

### 4.12. Enzyme-Linked Immunosorbent Assay (ELISA)

The cell culture supernatant from each group was collected, centrifuged, and then kept at −80 °C for future investigation. IL-6 and TNF-α concentrations in the supernatant were quantified using the respective ELISA kits in accordance with the manufacturer’s guidelines. The optical density of each well was assessed at the designated wavelength with an advanced microplate reader. The concentrations of IL-6 and TNF-α were determined using the standard curve.

### 4.13. Statistical Analysis

All data were statistically evaluated using GraphPad Prism software (version 9.3.1) and presented as mean ± SEM. An ordinary one-way ANOVA was used for data analysis. Distinct letters in the histograms indicate significant differences (*p* < 0.05).

## 5. Conclusions

This study elucidates the molecular connection between HMGB1-mediated inflammation and genes associated with ovulatory dysfunction, thereby establishing curcumin as a multi-target therapeutic drug for ovarian inflammatory disorders. The key findings demonstrate that HMGB1 activates *TLR2*-NF-κB signaling in BOGCs, driving IL-6 hypersecretion and dysregulation of ovulatory genes. Furthermore, curcumin suppresses *TLR2* expression and may interfere with HMGB1-*TLR2* binding, thereby synergistically inhibiting NF-κB nuclear translocation. Additionally, the activity of curcumin to normalize the expression of *EGFR*, *VEGF*, and *STAR* expression reveals its potential to simultaneously mitigate inflammatory responses and restore ovulatory function—a unique advantage that distinguishes it from conventional anti-inflammatory agents. Although the bovine model provides a valuable information system for studying ovarian physiology, interspecies differences in signaling pathways, hormonal regulation, and reproductive biology limit the applicability of these findings to other species. Significant distinctions across species may reside in the precise modulation of the *TLR2*-NF-κB pathway or the expression patterns of ovulation-related genes. Consequently, the efficacy of curcumin in addressing inflammation in human granulosa cells requires more validation.

## Figures and Tables

**Figure 1 ijms-26-09180-f001:**
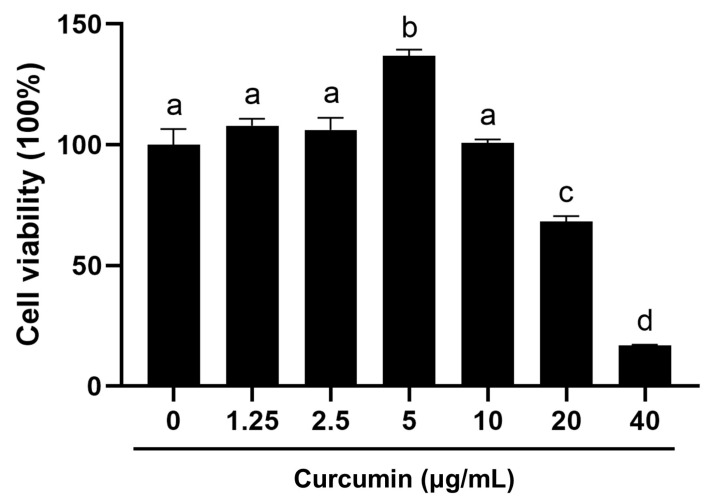
The effects of curcumin on BOGCs viability tested by MTT assay. Distinct lowercase letters in the figure denote statistically significant differences (*p* < 0.05).

**Figure 2 ijms-26-09180-f002:**
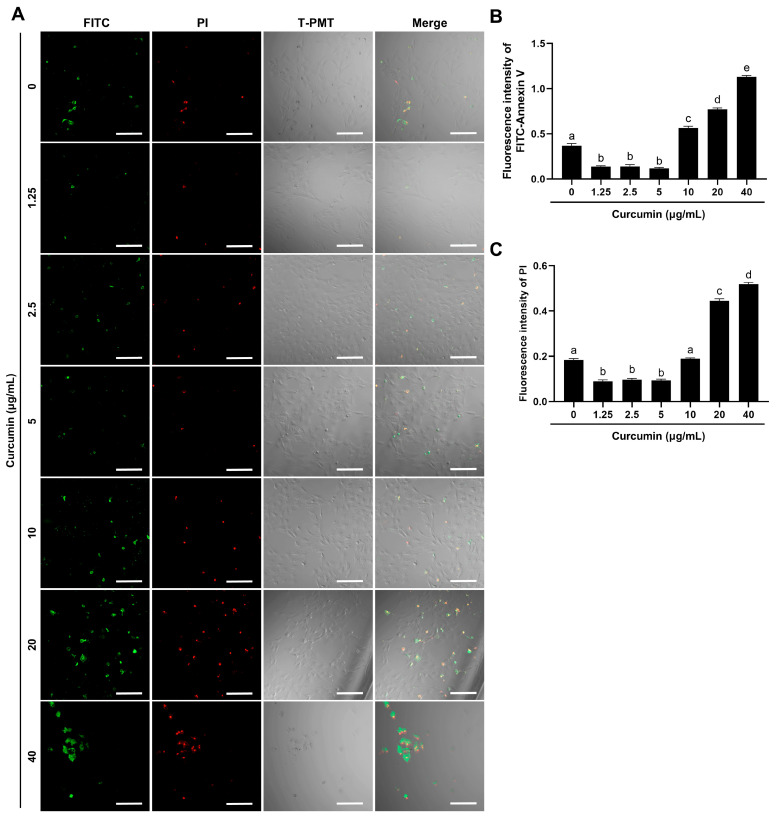
The effects of curcumin on apoptosis and necrosis of BOGCs. (**A**) Representative images of the FITC-Annexin V fluorescence probe, PI staining, and T-PMT (a transmitted detection module with laser confocal scanning microscope) in BOGCs; (**B**) Quantitative analysis of FITC-Annexin V fluorescence in BOGCs; (**C**) Quantitative analysis of PI staining in BOGCs. Green fluorescence indicated apoptotic cells labeled with FITC-Annexin V, whereas red fluorescence represented deceased cells labeled with PI. Bar: 100 μm. Distinct lowercase letters in the figure denote statistically significant differences (*p* < 0.05).

**Figure 3 ijms-26-09180-f003:**
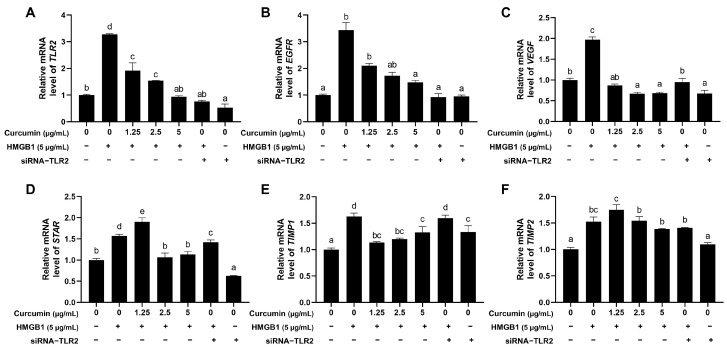
Curcumin reduced the effect of HMGB1 on the mRNA expression of target genes. (**A**–**F**) Relative mRNA expression levels of *TLR2* (**A**), *EGFR* (**B**), *VEGF* (**C**), *STAR* (**D**), *TIMP1* (**E**), and *TIMP2* (**F**). Distinct lowercase letters in the figure denote statistically significant differences (*p* < 0.05).

**Figure 4 ijms-26-09180-f004:**
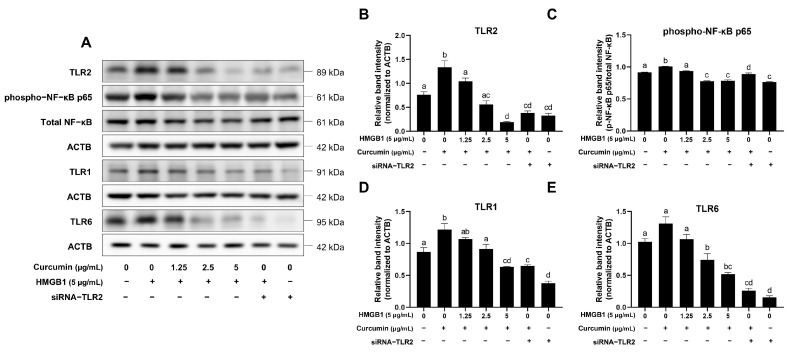
Curcumin inhibited protein expression of toll-like receptors and phospho-NF-κB p65 induced by HMGB1. (**A**) Immunoblots of *TLR2*, phospho-NF-κB p65, total NF-κB, *TLR1*, and *TLR6* in BOGCs; (**B**) Relative band intensity of *TLR2*/ACTB; (**C**) Relative band intensity of phospho-NF-κB p65/total NF-κB; (**D**) Relative band intensity of *TLR1*/ACTB; (**E**) Relative band intensity of *TLR6*/ACTB. ACTB as an internal control. Distinct lowercase letters in the figure denote statistically significant differences (*p* < 0.05).

**Figure 5 ijms-26-09180-f005:**
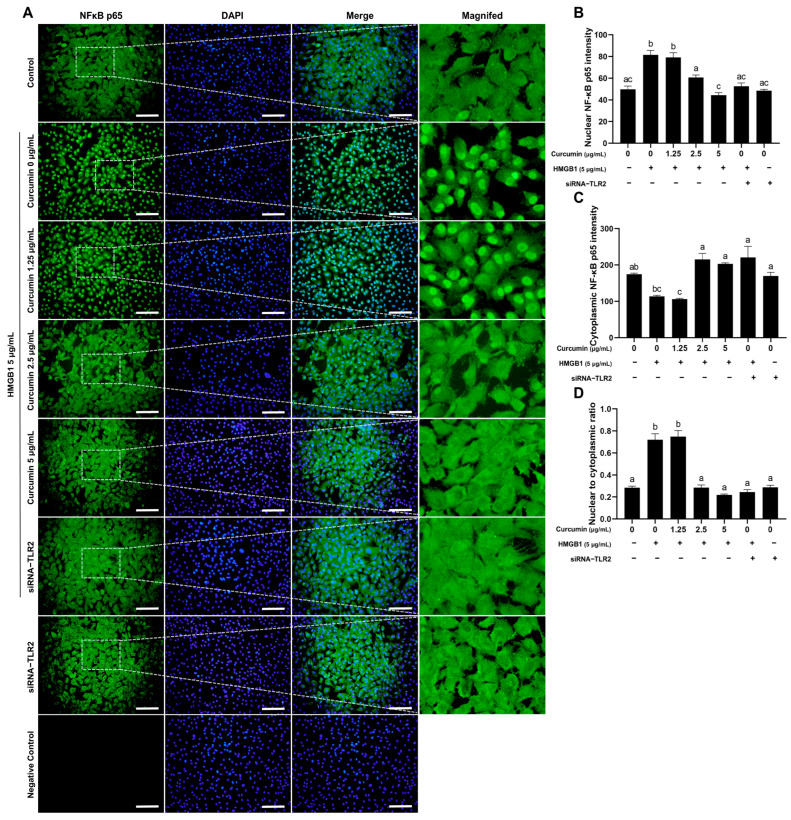
Curcumin inhibits HMGB1-induced nuclear translocation of NF-κB p65. (**A**) Representative images of NF-κB immunofluorescence and DAPI staining in BOGCs; (**B**,**C**) A quantitative examination of nuclear (**B**) and cytoplasmic (**C**) NF-κB p65 staining was performed; (**D**) A quantitative examination of the NF-κB p65 nuclear to cytoplasm ratio in BOGCs. NF-κB p65 exhibited green fluorescence when positive. Nuclei stained with DAPI exhibited blue fluorescence. Bar: 200 μm. Bar: 200 μm. Distinct lowercase letters in the figure denote statistically significant differences (*p* < 0.05).

**Figure 6 ijms-26-09180-f006:**
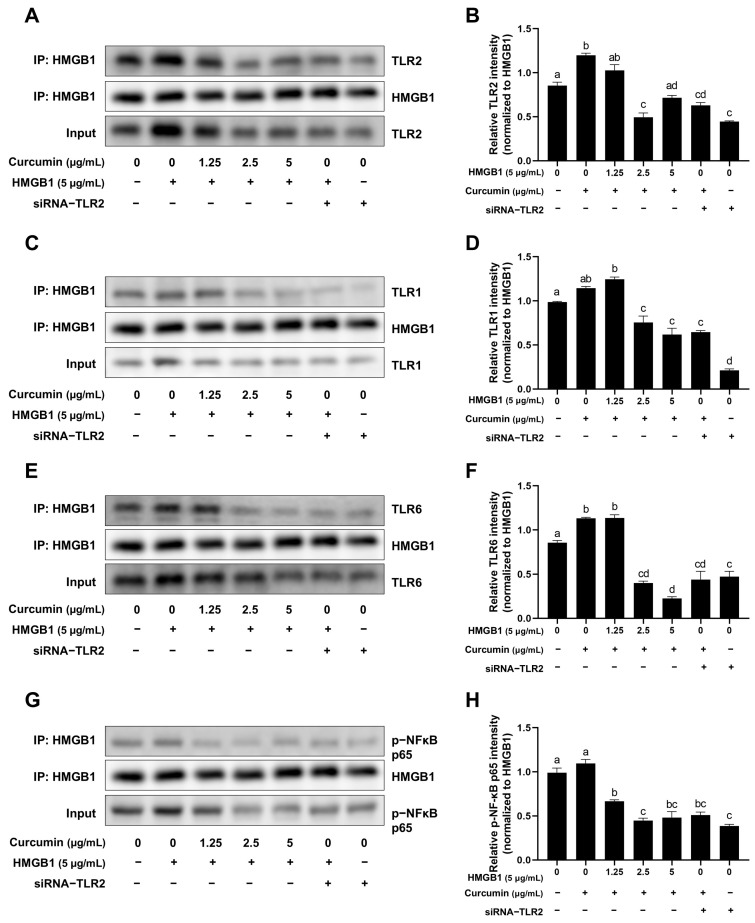
Curcumin attenuates the binding of HMGB1 to *TLR1*, *TLR2*, *TLR6*, and phospho-NF-κB p65. (**A**,**C**,**E**,**G**) Co-IP analysis of HMGB1 with *TLR2*, *TLR1*, *TLR6* and phospho-NF-κB p65 interaction. (**B**,**D**,**F**,**H**) Relative band intensity of *TLR2*, *TLR1*, *TLR6*, and phospho-NF-κB p65. Distinct lowercase letters in the figure denote statistically significant differences (*p* < 0.05).

**Figure 7 ijms-26-09180-f007:**
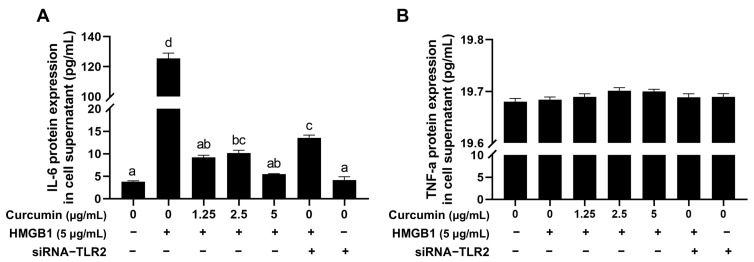
Curcumin reduces the HMGB1-induced increase of the inflammatory factor IL-6. (**A**) The concentration of IL-6 in the cellular supernatant. (**B**) The concentration of TNF-α in the cellular supernatant. Distinct lowercase letters in the figure denote statistically significant differences (*p* < 0.05).

**Table 1 ijms-26-09180-t001:** Primer information.

Gene Name	Primer Sequence (5′→3′)	Accession No.
*T* *LR* *2*	Forward: 5′-GCA CTT CAA CCC TCC CTT TA-3′ Reverse: 5′-GTT CTC CGA AAG CAC AAA GAT G-3′	NM_174197.2
*E* *GFR*	Forward: 5′-CTA TGA CCC TAC CAC CTA CGA-3′ Reverse: 5′-AAA CTC ACC GAT TCC TAT TCC-3′	XM_002696890.5
*V* *EGF*	Forward: 5′-CCC ACG AAG TGG TGA AGT TCA-3′ Reverse: 5′-CCA CCA GGG TCT CGA TGG-3′	NM_001316955.1
*S* *TAR*	Forward: 5′-GTG GAT TTT GCC AAT CAC CT-3′ Reverse: 5′-TTA TTG AAA ACG TGC CAC CA-3′	NM_174189.3
*T* *IMP* *1*	Forward: 5′-CCT ATG CTG CTG GTT GTG AGG A-3′ Reverse: 5′-AGT GAG TGT CGC TCT GCA GTT TG-3′	NM_174471.3
*T* *IMP* *2*	Forward: 5′-ACA GGT ACC AGA TGG GCT GTG AG-3′ Reverse: 5′-CGT GAC CCA GTC CAT CCA GA-3′	NM_174472.4
*ACTB*	Forward: 5′-ATC GGC AAT GAG CGG TTC C-3′ Reverse: 5′-GTG TTG GCG TAG AGG TCC TTG-3′	NM_173979.3

## Data Availability

Data are contained within the article and Appendix A.

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
