# Peer review of "Curcumin Alleviates HMGB1-Mediated Inflammation Through the Signaling Pathway of *TLR2*-NF-κB in Bovine Ovarian Granulosa Cells"

_ijms, 2025, doi:10.3390/ijms26189180_

Round 1

Reviewer 1 Report

Comments and Suggestions for Authors

The article titled Curcumin alleviates HMGB1-induced inflammation via the TLR2-NF-κB pathway in bovine ovarian granulosa cells” investigates the molecular mechanisms by which curcumin mitigates HMGB1-induced inflammation and ovulatory dysfunction in bovine ovarian granulosa cells (BOGCs), with a focus on the TLR2–NF-κB signaling pathway. The authors integrate multiple experimental approaches, including cell viability assays, apoptosis staining, RT-qPCR, Western blotting, immunofluorescence, and co-immunoprecipitation, to examine curcumin’s effects on inflammatory and ovulatory markers. The findings suggest that curcumin at lower concentrations (<5 μg/mL) exerts cytoprotective and anti-inflammatory effects, while higher concentrations become cytotoxic. The manuscript also highlights curcumin’s ability to disrupt HMGB1 receptor interactions and selectively suppress IL-6 expression without affecting TNF-α, suggesting a degree of pathway specificity. However, certain aspects of the methodology, data interpretation, and contextual discussion could be expanded or clarified to strengthen the manuscript's scientific depth and broader significance. The reviewer has the following comments that authors need to address:

  1. The rationale for curcumin concentration selection needs to be clearly justified, with reference to relevant literature and dose–response considerations. The authors should also provide a plausible explanation for the observed increase in cell viability at the 5 μg/mL dose.

  1. The manuscript discusses bioavailability limitations and briefly suggests nanoparticle encapsulation as a strategy. This section should be expanded to address pharmacokinetics, formulation strategies, and realistic challenges for clinical or veterinary application.

  1. The downstream functional consequences (e.g., ovulation rates, fertility outcomes) remain speculative. To strengthen translational relevance, the authors could consider complementing the current molecular data with functional studies, such as in vitro fertilization assays, hormone production analysis, or in vivo reproductive outcome models. Without such connections, the translational claims remain premature.

  1. The manuscript discusses bioavailability limitations and briefly suggests nanoparticle encapsulation as a strategy. This section should be expanded to address pharmacokinetics, formulation strategies, and realistic challenges for clinical or veterinary application.

  1. The manuscript would benefit from citing given recent work on diarylheptanoid derivatives, which provide important context for curcuminoid chemistry and their anti-inflammatory activities that would help readers better appreciate the structural and pharmacological basis of the present study.

https://aces.onlinelibrary.wiley.com/doi/full/10.1002/asia.202400380

  1. Interspecies variability and the relevance of bovine models to human ovarian biology should be critically addressed. While bovine granulosa cells are informative, key differences in signaling, hormonal regulation, and reproductive physiology limit direct extrapolation to humans. The authors should acknowledge these limitations, cite comparative studies, and suggest follow up validation in human granulosa cells or appropriate in vivo models.

Author Response

First of all, we are very grateful to the reviewers and editors for their expertise and opinion. To address the comments, we modified the manuscript text and figures and made thorough modifications to the language to improve clarity, fluency, and readability. We highlighted in red color on the changes made to the text in response to the reviewers' comments. We have tried our best to improve the manuscript. We provide a point-by-point responses to all of the comments.

Reviewer #1:

The article titled “Curcumin alleviates HMGB1-induced inflammation via the TLR2-NF-κB pathway in bovine ovarian granulosa cells” investigates the molecular mechanisms by which curcumin mitigates HMGB1-induced inflammation and ovulatory dysfunction in bovine ovarian granulosa cells (BOGCs), with a focus on the TLR2–NF-κB signaling pathway. The authors integrate multiple experimental approaches, including cell viability assays, apoptosis staining, RT-qPCR, Western blotting, immunofluorescence, and co-immunoprecipitation, to examine curcumin’s effects on inflammatory and ovulatory markers. The findings suggest that curcumin at lower concentrations (<5 μg/mL) exerts cytoprotective and anti-inflammatory effects, while higher concentrations become cytotoxic. The manuscript also highlights curcumin’s ability to disrupt HMGB1 receptor interactions and selectively suppress IL-6 expression without affecting TNF-α, suggesting a degree of pathway specificity. However, certain aspects of the methodology, data interpretation, and contextual discussion could be expanded or clarified to strengthen the manuscript's scientific depth and broader significance. The reviewer has the following comments that authors need to address:

  1. The rationale for curcumin concentration selection needs to be clearly justified, with reference to relevant literature and dose–response considerations. The authors should also provide a plausible explanation for the observed increase in cell viability at the 5 μg/mL dose.

Response: We agree with the reviewer comments. It is very important to provide a looking forward theoretical basis for concentration selection and to provide explanations for anomalies. Recent research indicates that curcumin diminishes cell viability in human and murine colon cancer cells [1], but treatment with carvacrol and curcumin-loaded nanoparticles enhances osteoblast vitality by 1.4-fold [2]. Furthermore, it has been shown that curcumin safeguards rat H9C2 cardiomyocytes from hypoxia/reoxygenation damage, exhibiting a biphasic effect [3], thus indicating that curcumin's stimulatory and inhibitory impacts on cell survival are contentious. Our study found that the increase in cell viability observed at a dose of 5 μg/mL may be because low concentrations of curcumin enhance cell viability by activating antioxidant and pro-survival signaling pathways (such as Nrf2 and AKT) [4, 5]. According to your suggestion, we have added relevant literature citations in the corresponding results section (lines 101-102).

  1. Zhou, H.; Zhuang, Y.; Liang, Y.; Chen, H.; Qiu, W.; Xu, H.; Zhou, H. Curcumin exerts anti-tumor activity in colorectal cancer via gut microbiota-mediated CD8(+) T Cell tumor infiltration and ferroptosis. Food Funct. 2025, 16, 3671-3693. [CrossRef]
  2. Dahiya, A.; Chaudhari, V.S.; Bose, S. Bone Healing via Carvacrol and Curcumin Nanoparticle on 3D Printed Scaffolds. Small2024, 20, e2405642. [CrossRef]
  3. Yuan, Y.; Huang, H.; Hu, T.; Zou, C.; Qiao, Y.; Fang, M.; Liu, J.; Lai, S. Curcumin pretreatment attenuates myocardial ischemia/reperfusion injury by inhibiting ferroptosis, autophagy and apoptosis via HES1. Int.J. Mol. Med. 2024, 54, 110. [CrossRef]
  4. Wu, X.; Zhou, X.; Lai, S.; Liu, J.; Qi, J. Curcumin activates Nrf2/HO-1 signaling to relieve diabetic cardiomyopathy injury by reducing ROS in vitro and in vivo. FASEBJ 2022, 36, e22505. [CrossRef]
  5. Ren, B.C.; Zhang, Y.F.; Liu, S.S.; Cheng, X.J.; Yang, X.; Cui, X.G.; Zhao, X.R.; Zhao, H.; Hao, M.F.; Li, M.D.; et al. Curcumin alleviates oxidative stress and inhibits apoptosis in diabetic cardiomyopathy via Sirt1-Foxo1 and PI3K-Akt signalling pathways. J.Cell. Mol. Med. 2020, 24, 12355-12367. [CrossRef]

  1. The manuscript discusses bioavailability limitations and briefly suggests nanoparticle encapsulation as a strategy. This section should be expanded to address pharmacokinetics, formulation strategies, and realistic challenges for clinical or veterinary application.

Response: We agree with the reviewer comments. A more in-depth discussion of the pharmacokinetic challenges and translational potential of curcumin is critical to the impact of our study. As your suggest, we have expanded the discussion section to address these issues in detail (lines 250-260, lines 264-268).

  1. The downstream functional consequences (e.g., ovulation rates, fertility outcomes) remain speculative. To strengthen translational relevance, the authors could consider complementing the current molecular data with functional studies, such as in vitro fertilization assays, hormone production analysis, or in vivo reproductive outcome models. Without such connections, the translational claims remain premature.

Response: We have modified the discussion section to strengthen the relevance of the translation based on your comments. We have deleted the over-translated parts and replaced phrases such as "hint at the potential for infertility treatment" with more cautious language (lines 197-198, lines 215-217, lines 431-435).

  1. The manuscript discusses bioavailability limitations and briefly suggests nanoparticle encapsulation as a strategy. This section should be expanded to address pharmacokinetics, formulation strategies, and realistic challenges for clinical or veterinary application.

Response: We agree with the reviewer comment. A more in-depth discussion of the pharmacokinetic challenges and translational potential of curcumin is critical to the impact of our study. As your suggest, we have expanded the discussion section to address these issues in detail (lines 250-260, lines 264-268).

  1. The manuscript would benefit from citing given recent work on diarylheptanoid derivatives, which provide important context for curcuminoid chemistry and their anti-inflammatory activities that would help readers better appreciate the structural and pharmacological basis of the present study. https://aces.onlinelibrary.wiley.com/doi/full/10.1002/asia.202400380

Response: Thank you very much! We have added the content of "Curcumin as a linear diarylheptanoid compound has a wide range of anti-inflammatory activity" in the introduction part of the revised manuscript and cited the literature you provided and other related key studies (lines 42-46).

  1. Interspecies variability and the relevance of bovine models to human ovarian biology should be critically addressed. While bovine granulosa cells are informative, key differences in signaling, hormonal regulation, and reproductive physiology limit direct extrapolation to humans. The authors should acknowledge these limitations, cite comparative studies, and suggest follow up validation in human granulosa cells or appropriate in vivo models.

Response: We agree with the reviewer comment that acknowledging the limitations of the bovine model is essential for accurately assessing the impact and translational relevance of our findings. We have thoroughly addressed this limitation in the Conclusion section and proposed subsequent validation using human granulosa cells or suitable in vivo models. (lines 436-442).

Reviewer 2 Report

Comments and Suggestions for Authors

Dear Authors,

your study provides novel and significant insights into the molecular mechanisms by which curcumin mitigates HMGB1-induced inflammation in bovine ovarian granulosa cells. Although curcumin is well recognized for its anti-inflammatory and antioxidant properties, its effects in inflammatory ovarian disorders, particularly on ovarian granulosa cells, remain incompletely understood. The experimental design is robust, and the results are relevant for advancing our understanding of ovarian inflammatory pathology. After careful evaluation, I recommend the manuscript for publication pending minor revisions to enhance clarity, consistency, and readability. Specific suggestions are outlined below:

  1. The abstract is informative but could be more concise. Consider simplifying long sentences to clearly present the background, objectives, key findings, and significance.

  2. Correct verb agreement errors (e.g., “curcumin inhibit” → “curcumin inhibits”).

  3. The abstract ends abruptly (“providing a theoretical”). Please complete the concluding sentence to emphasize the broader implications of your findings.

  4. In the Introduction, justify the selection of curcumin concentrations used in the study (1.25–2.5 and 10 μg/mL).

Concerning the results section:

  • Ensure consistent terminology: “granular cells” and “granulosa cells” are used interchangeably. Use one term throughout the manuscript.

  • Clarify the effects of different curcumin concentrations, specifying which endpoints showed no significant change.

  • Some statements convey interpretation rather than strictly reporting results (e.g., “indicating additional TLR2-independent mechanisms”). These should be moved to the Discussion.

  • Including an LDH assay to complement the MTT results would strengthen conclusions regarding cell viability.

  • Ensure consistent use of symbols in figures to indicate statistically significant differences.

  • Improve the quality of immunofluorescence images, particularly those in Figure 5.

The discussion is generally well written, but could be tightened to reduce redundancy, particularly in describing HMGB1’s role in ovarian inflammation.

  • Briefly address why IL-6, but not TNF-α, was affected, as this is a biologically relevant observation.
  • Correct minor grammatical issues (e.g., “This findings indicates” → “These findings indicate”).
  • Conduct a thorough language and style revision to improve clarity and readability. Simplifying overly complex sentences will enhance accessibility for an international audience.

  • Ensure consistency in gene/protein nomenclature (italicization, capitalization) throughout the manuscript.

Once these minor revisions are addressed, I believe the manuscript will be suitable for publication.

Comments on the Quality of English Language
  1. Sentence Structure and Clarity
    • Many sentences are overly long and complex, making the text harder to follow. Consider breaking them into shorter sentences to improve readability.
    • Some sentences mix multiple ideas, particularly in the Abstract and Results, which can obscure the main point. Example: “Further research shows that curcumin inhibits the binding of HMGB1 protein with TLR1, TLR2, TLR6, and phospho-NF-κB p65, thereby suppressing the expression of the pro-inflammatory cytokine IL-6 and alleviating HMGB1-induced inflammation in OGCs.” Could be simplified into two sentences: one for the binding effect and one for the downstream outcome.
  1. Verb Agreement and Tense:
    • Occasional verb agreement errors are present, e.g., “curcumin inhibit” → “curcumin inhibits.”
    • Ensure consistent use of past tense when describing results (e.g., “demonstrated,” “showed”) and present tense for established knowledge (e.g., “Curcumin is…”).
  2. Terminology:
    • Terms such as “granular cells” vs. “granulosa cells” are used interchangeably; this can confuse readers. Pick one term and use it consistently.
    • Some phrases are awkward or non-standard in academic English. For instance, “low-promoting and high-inhibiting characteristic” could be rephrased as “dose-dependent dual effect” or “biphasic effect.”
  3. Consistency of Gene and Protein Names
    • Gene/protein nomenclature is inconsistently formatted (italicization, capitalization). Standardize throughout according to journal guidelines.
  4. Minor Grammatical Issues
    • Examples:
      • “This findings indicates” → “These findings indicate.”
      • “the optimal inhibitory concentrations of gene, curcumin were…” → “the optimal inhibitory concentrations of curcumin for each gene were…”
    • Articles (a, an, the) are sometimes missing or misused.
  5. Abstract and Conclusions
    • The Abstract contains incomplete or abrupt sentences (“providing a theoretical”). Ensure all sentences are complete and convey a clear message.
    • Avoid overly technical or convoluted phrasing in the Abstract; aim for concise, accessible English suitable for an international readership.

The manuscript is understandable, but a thorough language revision is recommended to improve clarity, flow, and readability. A professional English editing service or careful proofreading by a native or highly proficient English speaker could address these issues efficiently.

Author Response

First of all, we are very grateful to the reviewers and editors for their expertise and opinion. To address the comments, we modified the manuscript text and figures and made thorough modifications to the language to improve clarity, fluency, and readability. We highlighted in red color on the changes made to the text in response to the reviewers' comments. We have tried our best to improve the manuscript. We provide a point-by-point responses to all of the comments.

Reviewer #2:

your study provides novel and significant insights into the molecular mechanisms by which curcumin mitigates HMGB1-induced inflammation in bovine ovarian granulosa cells. Although curcumin is well recognized for its anti-inflammatory and antioxidant properties, its effects in inflammatory ovarian disorders, particularly on ovarian granulosa cells, remain incompletely understood. The experimental design is robust, and the results are relevant for advancing our understanding of ovarian inflammatory pathology. After careful evaluation, I recommend the manuscript for publication pending minor revisions to enhance clarity, consistency, and readability. Specific suggestions are outlined below:

  1. The abstract is informative but could be more concise. Consider simplifying long sentences to clearly present the background, objectives, key findings, and significance.

Response: Thank you for this constructive comment. We have revised the abstract to make it more concise, as suggested. Long sentences have been simplified, and the flow of background, results, and significance has been improved (lines 17-33).

  1. Correct verb agreement errors (e.g., “curcumin inhibit” → “curcumin inhibits”).

Response: We are grateful for this remark. We have corrected them in the revised version of the manuscript.

  1. The abstract ends abruptly (“providing a theoretical”). Please complete the concluding sentence to emphasize the broader implications of your findings.

Response: Thank you very much! According to your suggestion, we rewrote the conclusion sentence of the abstract, aiming to further elaborate the theoretical basis of our research findings and their potential application value in the field of cattle breeding (lines 29-33).

  1. In the Introduction, justify the selection of curcumin concentrations used in the study (1.25–2.5 and 10 μg/mL).

Response: Thank you very much! We have revised and supplemented the introduction of the manuscript. The curcumin concentrations of 0, 1.25, 2.5, 5, 10, 20, and 40 μg/ml selected in this study were mainly based on previously published literature in similar cell models. The concentration gradient we set is designed to cover its known effective biological range and is used to explore its possible dose-dependent effects. MTT assay results indicated that, compared to the control group, cell viability showed no significant changes at concentrations of 1.25, 2.5, and 10 μg/mL, while an increase in cell viability was observed at 5 μg/mL. In contrast, significant reductions in cell viability were noted at concentrations of 20 and 40 μg/mL. These findings are consistent with the bidirectional effects reported in previous studies [1]. Subsequently, fluorescence microscopy coupled with Annexin V/PI dual staining revealed a dichotomous effect of curcumin on BOGC survival. Low-dose curcumin (1.25-5 μg/mL) reduced apoptosis, while high-dose curcumin (20-40 μg/mL) exacerbated cell death. Therefore, the concentrations of curcumin at 1.25, 2.5 and 5 μg/ml were selected in the subsequent experiments (lines 48-58).

  1. Yuan, Y.; Huang, H.; Hu, T.; Zou, C.; Qiao, Y.; Fang, M.; Liu, J.; Lai, S. Curcumin pretreatment attenuates myocardial ischemia/reperfusion injury by inhibiting ferroptosis, autophagy and apoptosis via HES1. Int.J. Mol. Med. 2024, 54, 110. [CrossRef]

Concerning the results section:

  1. Ensure consistent terminology: “granular cells” and “granulosa cells” are used interchangeably. Use one term throughout the manuscript.

Response: We are grateful for this remark. We have corrected this mistake in the revised version of the manuscript (such as line 150).

  1. Clarify the effects of different curcumin concentrations, specifying which endpoints showed no significant change.

Response: We appreciate the reviewer's comment regarding the clarification of curcumin's effects at different concentrations. As shown in Fig. 1, curcumin exhibited a dose-dependent influence on bovine ovarian granulosa cell metabolic activity. Specifically, treatment with 5 μg/mL curcumin significantly enhanced cell viability, while concentrations equal to or exceeding 20 μg/mL markedly reduced it. Most importantly, no significant changes in cellular metabolic activity were observed at low and intermediate concentrations (1.25, 2.5, and 10 μg/mL) compared to the control group. Annexin V/PI double staining revealed that low-dose curcumin (1.25–5 μg/mL) reduced cell apoptosis, whereas high-dose curcumin (10–40 μg/mL) exacerbated it. Based on these findings, concentrations of 1.25, 2.5, and 5 μg/mL were selected for subsequent experiments to evaluate the specific therapeutic effects of curcumin independent of alterations in cell viability.

  1. Some statements convey interpretation rather than strictly reporting results (e.g., “indicating additional TLR2-independent mechanisms”). These should be moved to the Discussion.

Response: We thank the reviewer for this constructive comment. We removed the parts that were not strictly reporting results and added them to the discussion section (lines 217-220).

  1. Including an LDH assay to complement the MTT results would strengthen conclusions regarding cell viability.

Response: We thank the reviewer for raising this valuable point. We fully agree that incorporating LDH assay data would provide a complementary measure of cytotoxicity and further strengthen the conclusions regarding cell viability. In the future study on this topic, we will add this parameter in our experiment, because this research is carriing out.

  1. Ensure consistent use of symbols in figures to indicate statistically significant differences.

Response: We are grateful for this remark and corrected it in the revised version of the manuscript (Figure 1, lines 108-109).

  1. Improve the quality of immunofluorescence images, particularly those in Figure 5.

Response: We greatly appreciate this comment. In response, we have regenerated the image files at 600 dpi and uploaded them to the system and the revised manuscript to ensure optimal clarity of the images. We sincerely apologize that we are unable to obtain higher-resolution images, as the capabilities of our imaging system do not allow for further significant improvement in resolution. The current images represent the best possible clarity achievable with our available equipment (Figures 1-7).

  1. The discussion is generally well written, but could be tightened to reduce redundancy, particularly in describing HMGB1’s role in ovarian inflammation.

Response: We sincerely thank the reviewer for this insightful comment. We agree that the discussion section could be more concise. As your suggest, we have thoroughly revised the Discussion to remove redundant statements, particularly those concerning the role of HMGB1 in ovarian inflammation (lines 203-212).

  1. Briefly address why IL-6, but not TNF-α, was affected, as this is a biologically relevant observation.

Response: We thank the reviewer for raising this insightful point. The differential regulation of IL-6 and TNF-α is indeed biologically significant. We provide the following potential explanations based on existing literature:

While both IL-6 and TNF-α are NF-κB target genes, their promoters are regulated by distinct sets of transcription factors. The IL-6 promoter contains binding sites for NF-κB and C/EBPβ, which are both potently activated by the HMGB1-TLR2 pathway in our model [1]. In contrast, TNF-α transcription is more predominantly and directly controlled by NF-κB [2]. It is plausible that the curcumin-mediated inhibition, while sufficient to block IL-6 induction, may not have fully suppressed NF-κB activity below the threshold required for TNF-α transcription, which might have a higher sensitivity to NF-κB or utilize alternative regulatory elements less affected by our intervention.

HMGB1 can signal through multiple receptors (TLR2, TLR4, RAGE). Although our study focused on TLR2, it is possible that HMGB1-induced TNF-α expression in BOGCs might involve a contribution from other receptors (TLR4) that are not effectively targeted by the siRNA-TLR2 knockdown or the curcumin treatment at the concentrations used [3, 4]. This could explain the persistent TNF-α levels observed.

  • Ren, Q.; Liu, Z.; Wu, L.; Yin, G.; Xie, X.; Kong, W.; Zhou, J.; Liu, S. C/EBPβ: The structure, regulation, and its roles in inflammation-related diseases. Pharmacother.2023, 169, 115938. [CrossRef]
  • Vallabhapurapu, S.; Karin, M. Regulation and function of NF-kappaB transcription factors in the immune system. Rev. Immunol.2009, 27, 693-733. [CrossRef]

[3] Zusso, M.; Lunardi, V.; Franceschini, D.; Pagetta, A.; Lo, R.; Stifani, S.; Frigo, A.C.; Giusti, P.; Moro, S. Ciprofloxacin and levofloxacin attenuate microglia inflammatory response via TLR4/NF-kB pathway. J. Neuroinflammation 2019, 16, 148. [CrossRef]

[4] Fitzgerald, K.A.; Kagan, J.C. Toll-like Receptors and the Control of Immunity. Cell 2020, 180, 1044-1066. [CrossRef]

  1. Correct minor grammatical issues (e.g., “This findings indicate”→“These findings indicate”).

Conduct a thorough language and style revision to improve clarity and readability. Simplifying overly complex sentences will enhance accessibility for an international audience.

Response: We are grateful for this remark. We have corrected many grammatical issues in the revised version of the manuscript.

  1. Ensure consistency in gene/protein nomenclature (italicization, capitalization) throughout the manuscript.

Response: We agree with the reviewer on the importance of consistent gene and protein nomenclature throughout the manuscript. We have now thoroughly reviewed the entire text and have standardized all gene and protein symbols according to international guidelines. Specifically, all gene symbols are now italicized in uppercase, while protein designations are in upright font.

Once these minor revisions are addressed, I believe the manuscript will be suitable for publication.

  1. Sentence Structure and Clarity

Many sentences are overly long and complex, making the text harder to follow. Consider breaking them into shorter sentences to improve readability.

Some sentences mix multiple ideas, particularly in the Abstract and Results, which can obscure the main point. Example: “Further research shows that curcumin inhibits the binding of HMGB1 protein with TLR1, TLR2, TLR6, and phospho-NF-κB p65, thereby suppressing the expression of the pro-inflammatory cytokine IL-6 and alleviating HMGB1-induced inflammation in OGCs.” Could be simplified into two sentences: one for the binding effect and one for the downstream outcome.

Response: We are grateful for this remark. We have corrected it in the revised version of the manuscript (lines 25-29).

  1. Verb Agreement and Tense:

Occasional verb agreement errors are present, e.g., “curcumin inhibit” → “curcumin inhibits.”

Ensure consistent use of past tense when describing results (e.g., “demonstrated,” “showed”) and present tense for established knowledge (e.g., “Curcumin is…”).

Response: We are grateful for this remark. We have corrected it in the revised version of the manuscript.

  1. Terminology:

Terms such as “granular cells” vs. “granulosa cells” are used interchangeably; this can confuse readers. Pick one term and use it consistently.

Some phrases are awkward or non-standard in academic English. For instance, “low-promoting and high-inhibiting characteristic” could be rephrased as “dose-dependent dual effect” or “biphasic effect.”

Response: We are grateful for this remark. We have corrected it in the revised version of the manuscript.

  1. Consistency of Gene and Protein Names

Gene/protein nomenclature is inconsistently formatted (italicization, capitalization). Standardize throughout according to journal guidelines.

Response: We agree with the reviewer on the importance of consistent gene and protein nomenclature throughout the manuscript. We have now thoroughly reviewed the entire text and have standardized all gene and protein symbols according to international guidelines. Specifically, all gene symbols are now italicized in uppercase, while protein designations are in upright font.

  1. Minor Grammatical Issues

Examples:

“This findings indicates” → “These findings indicate.”

“the optimal inhibitory concentrations of gene, curcumin were…” → “the optimal inhibitory concentrations of curcumin for each gene were…”

Articles (a, an, the) are sometimes missing or misused.

Response: We are grateful for this remark. We have corrected it in the revised version of the manuscript.

  1. Abstract and Conclusions

The Abstract contains incomplete or abrupt sentences (“providing a theoretical”). Ensure all sentences are complete and convey a clear message.

Avoid overly technical or convoluted phrasing in the Abstract; aim for concise, accessible English suitable for an international readership.

Response: We thank the reviewer for this constructive comment. We have now revised the abstract to make it more concise, as suggested. Long sentences have been simplified, and the flow of background, results, and significance has been improved. According to your suggestion, we rewrote the conclusion sentence of the abstract, aiming to further elaborate the theoretical basis of our research findings and their potential application value in the field of cattle breeding (lines 17-33).

The manuscript is understandable, but a thorough language revision is recommended to improve clarity, flow, and readability. A professional English editing service or careful proofreading by a native or highly proficient English speaker could address these issues efficiently.

Response: Than you very much for your suggestion. We have invited a proffessinal teacher to help us improving the English language in the revised manuscript.

Round 2

Reviewer 1 Report

Comments and Suggestions for Authors

The authors have thoroughly addressed the comments raised by the previous reviewers, demonstrating careful attention to detail and a commitment to improving the manuscript. The article, in its current form, meets the standards of quality and scientific rigor expected for publication.